# Disentangling Restrictive and Repetitive Behaviors and Social Impairments in Children and Adolescents with Gilles de la Tourette Syndrome and Autism Spectrum Disorder

**DOI:** 10.3390/brainsci10050308

**Published:** 2020-05-18

**Authors:** Mariangela Gulisano, Rita Barone, Salvatore Alaimo, Alfredo Ferro, Alfredo Pulvirenti, Lara Cirnigliaro, Selena Di Silvestre, Serena Martellino, Nicoletta Maugeri, Maria Chiara Milana, Miriam Scerbo, Renata Rizzo

**Affiliations:** 1Child and Adolescent Psychiatric Section, Department of Clinical and Experimental Medicine, Catania University, Via Santa Sofia 78, 95123 Catania, Italy; rbarone@unict.it (R.B.); laracirnigliaro92@gmail.com (L.C.); selena_disilvestre@yahoo.it (S.D.S.); serena.martellino@live.it (S.M.); nico_maugeri@hotmail.it (N.M.); mariachiara.milana@gmail.com (M.C.M.); mimiscerbo@gmail.com (M.S.); rerizzo@unict.it (R.R.); 2Bioinformatics Unit, Department of Clinical and Experimental Medicine, Catania University, Viale Andrea Doria 6, 95125 Catania, Italy; alaimos@gmail.com (S.A.); ferro@dmi.unict.it (A.F.); apulvirenti@dmi.unict.it (A.P.)

**Keywords:** autism spectrum disorder, Gilles de la Tourette, obsession, compulsion, social behavior, social impairment

## Abstract

Gilles de la Tourette syndrome (GTS) and autism spectrum disorder (ASD) are two neurodevelopmental disorders with male predominance, frequently comorbid, that share clinical and behavioral features. The incidence of ASD in patients affected by GTS was reported to be between 2.9% and 22.8%. We hypothesized that higher ASD rates among children affected by GTS previously reported may be due to difficulty in discriminating GTS sub-phenotypes from ASD, and the higher scores in the restrictive and repetitive behaviors in particular may represent at least a “false comorbidity”. We studied a large population of 720 children and adolescents affected by GTS (*n* = 400) and ASD (*n* = 320), recruited from a single center. Patients were all assessed with The Yale Global Tic Severity Rating Scale (YGTSS), The Autism Diagnostic Observation Schedule (ADOS), The Autism Diagnostic Interview Revised (ADI-R), The Children’s Yale–Brown Obsessive–Compulsive Scale (CY-BOCS), and The Children’s Yale–Brown Obsessive–Compulsive Scale for autism spectrum disorder (CY-BOCS ASD). Our results showed statistically significant differences in ADOS scores for social aspects between GTS with comorbid attention deficit hyperactivity disorder (ADHD) and obsessive–compulsive disorder (OCD) sub-phenotypes and ASD. No differences were present when we compared GTS with comorbid ASD sub-phenotype to ASD, while repetitive and restrictive behavior scores in ASD did not present statistical differences in the comparison with GTS and comorbid OCD and ASD sub-phenotypes. We also showed that CY-BOCS ASD could be a useful instrument to correctly identify OCD from ASD symptoms.

## 1. Introduction

Autism spectrum disorder (ASD) is a neurodevelopmental disorder with an onset in early childhood characterized by persistent deficits in social communication and social interaction across multiple contexts alongside restricted and repetitive patterns of behavior and interests or activities [1]. Social problems are the most characteristic and persistent components of the behavioral phenotype. Autism is 4.5 times more common in males than in females. The prevalence of ASD is approximately 1% in the general population [2].

Gilles de la Tourette syndrome (GTS) is a neurodevelopmental disorder characterized by motor tics and at least one vocal tic that occurs for more than 1 year, with an age of onset before 18 years [1]. It is also characterized by male predominance. In addition to sharing some symptomatology and sexual dysmorphism, these disorders are commonly comorbid [3]. 

This comorbidity might partly reflect common etiological mechanisms in children and adolescents affected by GTS, and the strong heritability of both ASD and GTS implies genetic etiology [4]; the Brainstorm consortium (2018) reported shared heritability in both disorders [5]. However, another study examining the shared genetic etiology through analysis of heritability among brain phenotypes found common genetic variation across a number of neuropsychological disorders and proposed a specific relationship between GTS and ASD [6]. A large body of literature has convincingly shown that many similarities are present in genetic factors, functional and structural brain characteristics, and cognitive profiles. In particular, recent studies have found similar overlapping alterations of functional connectivity in ASD and GTS patients [7]. Compulsive attachment to routines and stereotypical behaviors in ASD are associated with changes in the corticostriatal circuitry especially involving the caudate [8]. Likewise, compulsions observed in obsessive–compulsive disorder (OCD) were shown to be associated with alteration of the orbitofrontal circuitry and caudate nucleus dysfunction [9]. 

The incidence of ASD in patients affected by GTS was reported to be between 2.9% and 4.9% in two clinical populations, using DSM-5 criteria for the diagnoses of both GTS and ASD [10,11]. The first study [10] analyzed the data of 3500 patients from the Tourette Syndrome International Database Consortium Registry that comprised 83 active sites, while the second reported a small group (35 patients) of Iranian children and adolescents aged 6–18 years [11]. 

More recently Huisman-van Dijk et al. and Darrow et al. [12,13] studied two large clinical cohorts of GTS patients and their family members using two screening instruments, the Autism Quotient and the Social Responsiveness Scale (SRS), to characterize ASD symptoms and reported a higher incidence of “probable” ASD between 20% and 22.8%. 

Social communication and interaction challenges as well as the presence of functionally disabling restricted repetitive behaviors are characteristic of ASD. Symptoms such as obsessions, compulsive behaviors, echolalia and palilalia are common in both conditions [14]. 

Repetitive behaviors are observed in as many as 65% of patients with GTS and can be classified as “tic-like” or OCD-like symptoms according to clinical phenomenology [15]. Repetitive behaviors in ASD typically overlap with phenomena in GTS. It may be challenging to distinguish phenomenological characteristics of ASD from GTS. In clinical practice, medical professionals often struggle to define which disorder best describes the child’s symptoms. 

Our hypothesis is that higher ASD rates among children affected by GTS reported by Darrow et al. [13] and Huiss Van Dick et al. [12] may be due to the difficulty in discriminating complex tics and OCD symptoms from ASD symptoms, and the higher scores in the repetitive behaviors in particular may represent at least a “false comorbidity”. 

The primary aim of this study was to disentangle repetitive behavior features and social impairment in two large populations from a single center, GTS patients compared to ASD patients. Specifically, we aimed to determine whether features related to autism as measured by ADOS subscales and CY-BOCS differ for GTS sub-phenotype patients in relation to ASD. 

## 2. Materials and Methods

### 2.1. Patient Selection

Participants were recruited from a larger database on clinical and molecular features of GTS and ASD that was assembled over a twenty-year period and is still being updated. 

Participants in the present study were 720 children and adolescents with a primary diagnosis of GTS or ASD, according to DSM-5 criteria (APA 2013), recruited from January 2016 to February 2019 at the outpatient clinic of the Child and Adolescent Neuropsychiatry Unit at Catania University Hospital. We excluded patients who showed 1) evidence of primary psychiatric disorders different from GTS or ASD and 2) severe neurological or physical impairments or who were minimally verbal.

### 2.2. Procedures

The study was approved by the local Ethics Committee. Investigations were carried out as part of the routine clinical care of the patients in accordance with the ethical standards laid down in the 1964 Declaration of Helsinki and its later amendments (Helsinki Declaration 1975, revision 2013). All parents gave written informed consent, and the subjects assented when possible. All patients underwent physical examination and blood and urine analyses to rule out systemic diseases (e.g., inherited metabolic disorders). Diagnoses of GTS and comorbid disorders and ASD were performed according to DSM5 criteria, and neuropsychological evaluation was carried out by an expert team of child and adolescent neurologists, supervised by R.R., M.G., and R.B. 

### 2.3. Measures 

All parents participated in a medical history interview to determine prenatal, perinatal, and psychosocial risk factors as well as socio-economic status. Before inclusion in the study, all patients were screened with the Schedule for affective disorders and Schizophrenia for School age children—present and lifetime (Kiddie-SADS-PL) to rule out primary psychiatric disorders considered as criteria of exclusion. The Kiddie-SADS-PL is a semi-structured interview tool developed by Kauffman et al. (1997) that can be used in children and adolescents aged between 6–18 years [16]. This evaluation includes fundamental diagnosis of various psychiatric disorders. All patients underwent neuropsychological evaluation for GTS and related comorbidities.

#### 2.3.1. Tics

To assess tic disorders, the National Hospital Interview Schedule for GTS (NHIS) [17], a semi-structured interview was used to diagnose GTS and its associated conditions, behaviors, and relevant family history. The Yale Global Tic Severity Rating Scale (YGTSS) [18] is an 11-item clinician rated interview that is able to evaluate motor and phonic tic severity considering the number, frequency, and impairments that tics are able to provoke in the patient. The score of the YGTSS ranges from 0 to 100, including the impairment section. Higher scores indicate greater severity of symptoms and impairment. 

#### 2.3.2. Obsessive–Compulsive Disorder

To evaluate OCD, the Children’s Yale–Brown Obsessive–Compulsive Scale (CY-BOCS) [19] and the ASD CY-BOCS [20] were used. The CY-BOCS is a semi-structured interview conducted principally with parents, even if patients are encouraged to participate. This interview is able to assess the severity of obsessive–compulsive symptoms in children. The total score of CY-BOCS ranges between 0 and 40. It is possible to evaluate an obsession and a compulsion score separately. Higher scores indicate greater severity of symptoms and impairment. 

#### 2.3.3. Autism Spectrum Disorder

The Autism Diagnostic Observation Schedule (ADOS) [21] was used for ASD diagnosis. The ADOS is a standardized evaluation scale that is considered the gold standard for the diagnosis of ASD. It is structured in four domains of exploration (A), social interaction (B), imagination (C), and repetitive and stereotyped behaviors (D). Items are scored from 0 to 3 on the basis of severity (3: more severe). The ADOS provides a total score and partial scores related to Social Affect (SA) and Restricted, Repetitive Behavior (RRB). 

The Autism Diagnostic Interview Revised (ADI-R) [22] is a structured interview conducted with the parents/guardians. Its purpose is to investigate the same areas of ADOS based on the judgment of parents/guardians with respect to their children’s symptoms. Items are scored from 0 to 3 on the basis of severity (3: more severe). 

The Children Yale Brown Obsessive–Compulsive Scale for Autism Spectrum Disorder is a semi-structured interview conducted with patients and parents that was developed after CY-BOCS by Schahill et al. (2014) to detect the differences between obsession and compulsion in OCD and repetitive and restrictive behaviors, interests, and stereotypes in ASD [20]. For this purpose, the scale is structured with a Symptom Checklist (behavior present or absent) and five sub-schedules (Time Spent, Interference, Distress, Resistance, and Control). Each item ranges from 0 (not present) to 4 (severe), and the total score is between 0 and 20. The interview includes a symptom checklist of possible repetitive behaviors grouped into eight categories (e.g., washing rituals, checking behaviors). Each category allows the interviewer to describe “other” behaviors presumed relevant to that overall category. The severity of each item is anchored to the past week and scored from 0 (not present) to 4 (severe) for a total score of 0 to 20 [20].

### 2.4. Statistical Analysis

Clinical covariates were identified within each psycho-diagnostic class that might be able to detect affected subjects and their comorbidities from healthy individuals. Analyzed psycho-diagnostic classes included ADI-R, ADOS, YGTSS, CY-BOCS, and CY-BOCS ASD.

The identification of class-specific clinical covariates, which have statistically different values, is a fundamental process to identify common elements among different conditions and to stratify affected subjects according to comorbidities. For this purpose, we employed the R software suite to perform statistical analyses using a two-sided t-test or ANOVA and to build a risk classification model based on C-Tree Induction [23].

Risk models were developed with the use of decision-tree induction from class-labeled training records, i.e., the training set was composed of records in which one attribute was the class label (or dependent variable), and the remaining attributes were the predictor variables; the individual records were the tuples for which the class label was known. 

The aim of this analysis was the objective identification of the classification power of evaluation metrics used in the context of Gilles de la Tourette syndrome and the assessment of a joint model to distinguish between pure GTS, GTS plus comorbidities, and ASD. For this purpose, the usage of a classification model is crucial to study the predictive power of a set of diagnostic metrics. The classification, starting from the characteristics (in our case the independent variables are those reported in Table 1), named predictor variables, of the examined subjects, aims to infer a set of rules allowing the prediction of covariate values, named class labels (or dependent variables). Each node of the tree analyzes a covariate by computing an optimal binary split. The goodness of the split is supported by a two-sample statistical test based on a permutation analysis.

In our study the objective was to distinguish six different classes: GTS, GTS + OCD, GTS + attention deficit hyperactivity disorder (ADHD), GTS + OCD + ADHD, GTS + ASD, and ASD.

The performance of the classification model was analyzed, randomly splitting the data in the training set (2/3 of the records) and the testing Set (1/3 of the records); this partition was repeated 100 times, and the classification accuracy was taken as the average of each accuracy measurement. All differences were considered statistically significant at a 5% probability level.

## 3. Results

We recruited a clinical cohort of 720 patients aged 6–18 years, mean age 11.3 (±3.3) years. Of this cohort, 400 subjects were affected by GTS. They were 274 males and 62 females, with a mean age of 11.4 (±3.1) years. Patients with ASD (*n* = 320) included 196 males and 37 females, with a mean age of 11.1 (±3.5). Demographic and clinical characteristics of the study samples are reported in Table 1. The GTS clinical cohort had a significantly higher mean IQ (91.7 ± 17.9) than that of the ASD patients (77.2 ± 24.4) (Table 1). 

Nearly all subjects with GTS presented at least one comorbidity. The results of the neuropsychological evaluation enabled assessment of GTS clinical sub-phenotypes distributed as follows: GTS only (15.5%), GTS + ADHD (32%), GTS + OCD (33.5%), GTS + ADHD + OCD (10.25%), and GTS + ASD (8.9%). GTS sub-phenotypes were compared to the ASD clinical group.

### 3.1. Yale Global Tic Severity Rating Scale

GTS patients presented a mean YGTSS total score of 19.5 (±9.4). The YGTSS score in the GTS clinical sub-phenotypes was as follows: pure GTS, 16.1 (±7); GTS+ADHD, 19.1 (±1); GTS+OCD, 22.2 (±9.7); GTS + ADHD + OCD, 25.9 (±9); and GTS + ASD, 11.7 (±3.2). The ASD group presented a mean YGTSS of 2.3 (±1.8). Statistically significant differences were observed in the comparisons of YGTSS scores between the GTS sub-phenotypes and ASD clinical cohort (*p* < 0.001) (see Figure 1 and Table 2).

### 3.2. Children’s Yale–Brown Obsessive–Compulsive Scale

GTS patients presented a mean CY-BOCS of 15.6 (±1). With regard to GTS clinical sub-phenotypes, the following mean CY-BOCS scores were computed: pure GTS, 7.6 (±6); GTS + ADHD, 9.9 (±7.8); GTS + OCD, 23.5 (±8.3); GTS + ADHD + OCD, 27.1 (±1); and GTS + ASD, 12.1 (±6.1). ASD patients presented a mean CY-BOCS score of 17.9 (±3.1) (Figure 2, Table 3).

### 3.3. Children’s Yale–Brown Obsessive–Compulsive Scale: Autism Spectrum Disorder

GTS patients presented a mean CY-BOCS ASD of 6.2 (±4.4). In detail, GTS clinical sub-phenotypes presented the following mean scores: pure GTS, 2.2 (±1.9); GTS + ADHD, 2.9 (±1.9); GTS + OCD, 6.2 (±4.4); GTS + ADHD + OCD, 3.1 (±2.1); and GTS + ASD, 17.9 (±2.9). ASD patients presented a mean CY-BOCS ASD score of 18.1 (±3.2) (Figure 2, Table 3).

Figure 3 shows a decision tree. Among all the inferred models, we showed the one built as a function of the CY-BOCS ASD variable. Each internal node of the tree represents a statistical test (on such a variable, the two departing branches split the data according to the most discriminant threshold). These analyses revealed that when the score of CY-BOCS ASD was lower than 1, the analyses were able to detect GTS only, whereas when the CY-BOCS ASD score was higher than 14, the more represented clinical group was that with pure ASD, followed by GTS+ASD. However, the overall accuracy of the model is 46,35%, which means that this variable is not reliable for detection of GTS + comorbidities when its value is between 1 and 14.

### 3.4. Autism Diagnostic Observation Schedule 

For all ADOS scales, ASD and GTS + ASD presented scores that fulfilled the diagnosis of ASD. No significant differences in scores were observed between these groups. In contrast, GTS presented significantly lower mean scores in ADOS scales compared to those of ASD and GTS + ASD. 

With regard to ADOS-SA “Social Interaction”, “Imagination”, and “Communication, Language, and Behavior”, the GTS sub-phenotypes always presented significantly lower scores than ASD and GTS + ASD sub-phenotypes. When ADOS-RR subscale scores of “restricted and repetitive behaviors” from the GTS sub-phenotype were analyzed in detail, different results related to the compared groups in each ADOS subscale were computed. In particular, no significant differences were detected when comparing GTS sub-phenotypes and ASD as follows: GTS + ASD vs. GTS + OCD in D3 (repetitive and stereotyped behavior) and D4 (compulsions); GTS + ASD vs. GTS + OCD + ADHD in D4 (compulsion); ASD vs. GTS + OCD in D1 (unusual interests), D3 (repetitive and stereotyped behavior), and D4 (compulsions) (Figure 4 and Table 4).

### 3.5. Autism Diagnostic Interview Revised

For all ADI-R scales, ASD and GTS + ASD presented scores that fulfilled the diagnosis of ASD. No significant differences in ADI-R scores were found between these groups. Pure GTS ADI-R scores did not fulfill the diagnosis of ASD and were significantly lower compared to those of ASD and GTS + ASD. Regarding the ADI-R scale “Restricted and Repetitive Behaviors” (C scale), no significant differences were detected when comparing GTS sub-groups and ASD as follows: GTS + ASD vs. GTS + OCD in C1 (unusual preoccupations), C2 (rituals), C3 (mannerism), and C4 (unusual interests); GTS + OCD vs. ASD in C2 (rituals) (Figure 5 and Table 5).

## 4. Discussion

Along with a deficit in social communication and interaction, restricted and repetitive behaviors constitute the defining core features of ASD; however, GTS also shares RRB and social communication and interaction problems with ASD.

In this study, we explored the prevalence and pattern of ASD symptoms in relation to GTS and other comorbid disorders, comparing the two distinct cohorts of individuals affected by GTS and ASD. We systematically assessed a large sample of children and adolescents affected by GTS and compared them to an ASD group of children and adolescents with verbal capacity.

As our primary hypothesis was that the higher percentage of ASD in GTS recently reported by other authors [12,13] showed a false comorbidity due to overlap between symptoms that were in common in both conditions, the study examined the degree to which the gold standard instruments for autism diagnosis, the ADI-R and ADOS, disentangle restrictive and repetitive behaviors and social impairment in children and adolescents affected by GTS and ASD and verify the hypothesis that, in particular, the presence of GTS + OCD and related symptoms could mimic ASD symptoms. The patients were also assessed by CY-BOCS and CY-BOCS ASD to investigate the overlap between ASD symptoms and GTS in affected individuals.

The results supported these hypotheses by indicating that higher ASD rates reported in other samples may be due to the confounding of tic or OCD symptoms for ASD or vice versa. In our samples of children and adolescents, 8.9% of patients affected by GTS presented comorbid ASD, in contrast with the incidences of 20% and 28% reported by Darrow et al. and Huisman-van Dijk et al., respectively [12,13].

In particular, when we compared ADOS and ADI-R with GTS and ASD, for ADOS scores, we did not observe any significant differences between GTS + ASD and GTS + OCD in D3 (repetitive and stereotyped behavior) and D4 (compulsions); GTS + ASD and GTS + OCD + ADHD in D4 (compulsion); or ASD and GTS + OCD in D1 (unusual interests), D3 (repetitive and stereotyped behavior), and D4 (compulsions). When examining ADI-R, no significant differences were detected between GTS + ASD and GTS + OCD in C1 (unusual preoccupations), C2 (rituals), C3 (mannerism), and C4 (unusual interests); GTS + ASD and GTS + OCD + ADHD in C1 (unusual preoccupations); and GTS + OCD and ASD in C2 (rituals).

Darrow et al. (2017) studied 535 GTS patients and used the Social Responsiveness Scale Second Edition (SRS) as a measure of ASD symptoms. The SRS contains five treatment subscales including the following domains: social awareness, social cognition, social communication (SC), social interaction (SCI), and restricted interests and repetitive behaviors (RRB).

When they examined the relationship between SRS total score and GTS sub-phenotypes, there were significant differences in SRS scores among the different classes. The highest SRS scores were found in the two classes that endorsed OCD symptoms (GTS + OCD + ADHD, OCD symmetry). These two classes had significantly higher SC and RRB scores than the other classes.

Our findings show that GTS + OCD and GTS + ASD sub-phenotypes were in line with this work; however, they reported a higher percentage (23%) of GTS-affected participants that met the cut-off criteria for probable ASD (83% of whom also met criteria for OCD). In our opinion this is because, as stated in previous studies, children [24] with mood and anxiety disorders have elevated rates of ASD based on SRS cut-off criteria, suggesting that some of the elevation in SRS scores may reflect underlying psychiatric impairment rather than being specific to ASD.

The fact that scores for the RRB subscales were higher for individuals who had OCD symptoms suggests that these SRS subscales may in fact be tapping into common repetitive behaviors in individuals with GTS and/or OCD that could be confused with stereotypies seen in ASD [13]

This is also supported by Huisman-van Dijk et al. (2016) in their significant paper, where items measuring repetitive behaviors in autism were loaded into a factor with OCD related items rather than into a factor with social communication items. Interestingly, the autism factor including lack of social skills as well as non-functional child routines, attention switching problems, and lack of investigation was not related to any of the tic or OCD symptom factors. This autism dimension might be etiologically distinct from the second factor characterized by repetitive behaviors [12].

In our study, the differences between GTS + OCD, GTS + ASD, and ASD samples were significant except in the ADOS and ADI-R domains of restrictive and repetitive behaviors.

More recently, Eapen et al. [25] in a sample of 203 participants, 44 with GTS and 26 with ASD compared to the general sample of 133 with a mean age of 18.17 years, examined the occurrence of autism related features measured by the Social Communication Questionnaire (SCQ) focusing on areas of overlap and differences.

They found that the GTS sample had significantly higher mean SCQ scores than the general population but generally lower scores than the ASD sample. The group differences in mean SCQ scores between GTS and ASD samples were significant except in the domain of RRB.

Eapen et al. suggested that symptom overlap may represent a true overlap through a shared phenotype or that the overlap may be a phenocopy where the clinical symptoms, for example, complex tics and related obsessive–compulsive symptoms of GTS, may mimic ASD symptoms and vice versa where stereotypic and repetitive behaviors characteristic of ASD may mimic GTS. In their study on the domain level, they found a close concordance between the two disorders on restricted and repetitive behaviors and less on social communication.

These results are in line with our study. ADOS differences between GTS clinical sub-phenotypes and ASD were significant in social interaction, imagination, and communication language and behavior; when RRB was analyzed, there were no significant differences.

However, restricted and repetitive behaviors are some of the major impairments shown in ASD and GTS. Having a measure that disentangles these two domains easily in a clinical setting is important for the clinicians who may use it as a basis for their treatment recommendations. We also assessed our clinical sample with CY-BOCS and CY-BOCS ASD.

CY-BOCS had the unique ability to detect symptomatology that differed from ASD characteristics, whereas other measures assessing obsessive–compulsive symptoms did not [26].

Several original CY-BOCS checklist items are nevertheless irrelevant for children with ASD. However, the results of this study provide an incremental refinement for measuring repetitive behavior in children and adolescents with GTS and ASD. Nonetheless, certain repetitive behaviors in children with ASD may have been overlooked due to sample characteristics of children with verbal capacity and normal or mild intellectual disability. Lower-functioning or non-verbal children were more likely to engage in hand and finger stereotypy and object manipulation. With regard to obsession and compulsion measured with CY-BOCS, no significant differences were detected in the comparison of scores in ASD, GTS + OCD, and GTS + ASD patients in our sample.

CY-BOCS ASD results provided a reliable and valid measure of repetitive behavior in youths with ASD and underscored differences in repetitive behavior in ASD compared to GTS [20]. With CY-BOCS ASD, no significant differences were found between GTS + ASD and ASD groups. The score obtained in the GTS + OCD, compared to the ASD group and GTS +ASD, was significantly lower. The clinical implications of these findings are important for clinicians who may use the CY-BOCS ASD as a diagnostic tool. The CY-BOCS ASD yields a total score based on the impact of the symptoms present rather than subscale scores. The results suggest that the CY-BOCS ASD is a valid and reliable measure of repetitive behavior in youths with ASD [27]. It is easy to use in a clinical setting, and clinicians may use it as a basis for their treatment recommendations.

### Limitations

The principal limitation of our study is that our Tourette Clinic is a tertiary level center. For this reason, we assessed and followed complicated cases that may have not been representative of the general population affected. The different sex distribution is a limitation of this sample because we analyzed two disorders with a high prevalence in males. Finally, another limitation was related to GTS sub-phenotype analysis and the comparison of GTS sub-phenotypes with ASD without comorbidities.

## 5. Conclusions

To conclude, the similarities largely contained within the RRS domain rather than across the RRS and social communication domains support our hypothesis that higher ASD rates among children affected by GTS may make it difficult to discriminate between specific features of GTS and ASD, contributing to the representation of a false comorbidity.

However, another possibility exists that certain symptoms may be duplicated due to environmental or other factors rather than due to a common genetic basis. The similarities could be due to a phenocopy rather than phenotypic overlap as suggested by Eapen et al. [25].

Genetic studies together with clinical task-based RMN functional studies could contribute to a better understanding of the underlying mechanism involved in the etiology of ASD and GTS and could be critical in obtaining better treatments.

Compared to their peers with GTS but not ASD, children with GTS and ASD have greater treatment needs (that are either unmet needs or treatment usage). This finding aligns with previous literature focused on children with ASD, which found high rates of service needs, particularly among children diagnosed with co-occurring conditions [28].

When symptoms are not correctly managed, more severe psychopathology may develop, alongside poorer interpersonal, school, family, and cognitive functioning and life outcomes among this population relative to that of controls. The possibility of individualizing symptoms to treat (e.g., compulsions, rituals, and repetitive behaviors) could improve the children’s quality of life and, consequently, other social aspects of their life. To our knowledge, this is the first study on a large unique cohort that analyzed symptoms that could mimic diagnosis. ADOS and ADI-R are the gold standard scales for diagnosis, and they are always able to individualize diagnosis. However, it might be helpful to define differences that could individualize treatment and help clinicians in the management of specific patients with GTS and comorbid OCD. In line with previous research [20], we suggest refining ASD diagnosis by using CY-BOSC ASD to better distinguish repetitive behaviors from compulsion and rituals.

## Figures and Tables

**Figure 1 brainsci-10-00308-f001:**
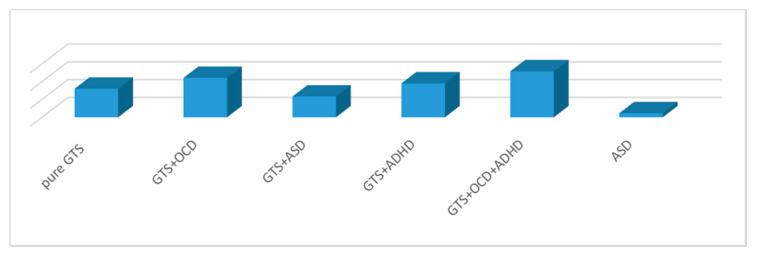
YGTSS score between clinical groups. GTS: Gilles de la Tourette Syndrome; ASD: Autism Spectrum Disorder; OCD: Obsessive–Compulsive Disorder; ADHD: Attention Deficit Hyperactivity Disorder; YGTSS: Yale Global Tic Severity Scale.

**Figure 2 brainsci-10-00308-f002:**
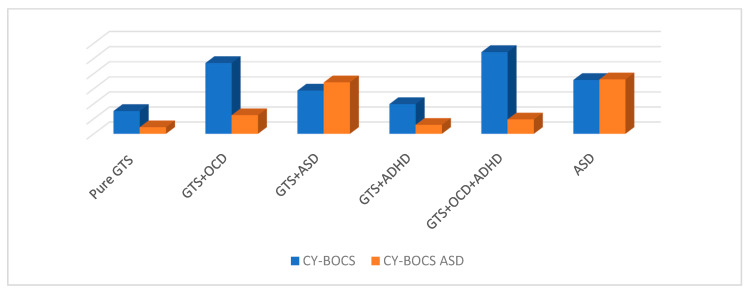
Comparison between CY-BOCS and CY-BOCS ASD scores between clinical groups. GTS: Gilles de la Tourette Syndrome; ASD: Autism Spectrum Disorder; OCD: Obsessive–Compulsive Disorder; ADHD: Attention Deficit Hyperactivity Disorder; CY-BOCS: Children Yale Brown Obsessive–Compulsive Scale; CY-BOCS ASD: Children Yale Brown Obsessive–Compulsive Scale for Autism Spectrum Disorder.

**Figure 3 brainsci-10-00308-f003:**
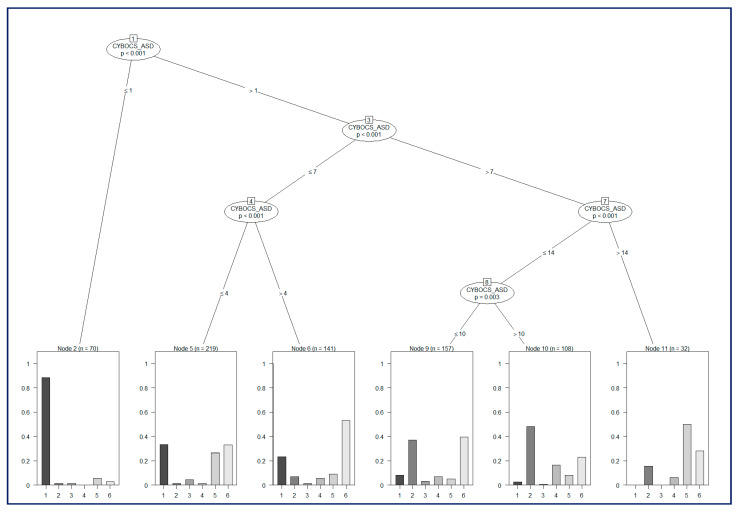
Decision tree. Clinical groups: 1: pure GTS; 2: GTS+ OCD; 3: GTS + ADHD; 4: GTS + ADHD + OCD + OCD; 5: GTS + ASD; 6 ASD.

**Figure 4 brainsci-10-00308-f004:**
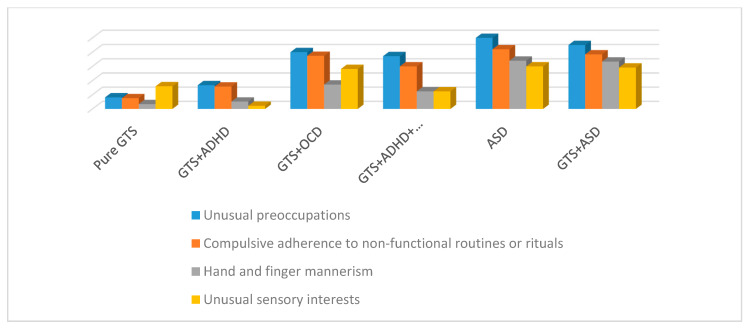
ADOS (D scale) restricted and repetitive behaviors: comparison between clinical sub-phenotypes. GTS: Gilles de la Tourette Syndrome; ASD: Autism Spectrum Disorder; OCD: Obsessive–Compulsive Disorder; ADHD: Attention Deficit Hyperactivity Disorder.

**Figure 5 brainsci-10-00308-f005:**
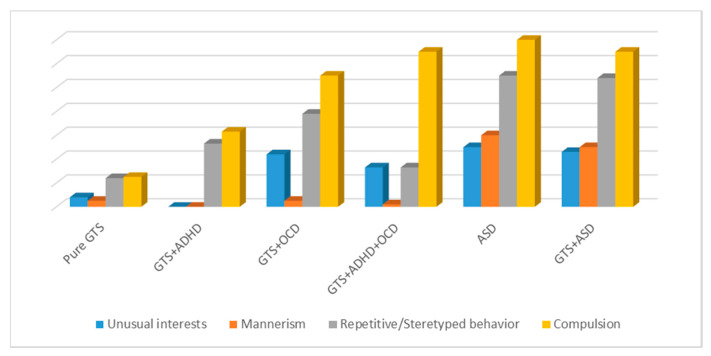
ADI-R, Restricted and Repetitive Behaviors (C Scale): comparison between clinical sub-phenotypes.

**Table 1 brainsci-10-00308-t001:** Demographic and clinical features.

	GTS (pt 400)	ASD (pt 320)
M/F	274/62	196/37
Mean age	11.4 (± 3.1)	11.1 (± 3.5)
IQ		
TIQ	91.7 (± 17.9)	77.2 (± 24.4)
PIQ	91.61 (± 18.0)	79.1 (± 24)
VIQ	92.87 (± 18.5)	77.7 (± 25.1)
YGTSS	19.54 (± 9.4)	1.4 (± 2.6)
CY-BOCS	15.61 (± 10.9)	13,9 (± 11,3)
CY-BOCS ASD	6.2 (± 4.4)	17 (± 3.2)

GTS: Gilles de la Tourette Syndrome; ASD: Autism Spectrum Disorder; CY-BOCS: Children Yale Brown Obsessive–Compulsive Scale; CY-BOCS ASD: Children Yale Brown Obsessive–Compulsive Scale for Autism Spectrum Disorder; YGTSS: Yale Global Tic Severity Scale; TIQ: Total Intelligence Quotient; PIQ: Performance Intelligence Quotient; VIQ: Verbal Intelligence Quotient.

**Table 2 brainsci-10-00308-t002:** Comparison between mean YGTSS values in each sub-phenotype and statistical significance.

	GTS only	GTS + OCD	GTS + ASD	GTS + ADHD	GTS + OCD + ADHD	ASD	*p*-Value
**YGTSS**	16.1 (±7)	22.2 (±9.7)	11.7 (3.2)	19.1 (±1)	25.9 (±9.6)	2.3 (±108)	0.0005

GTS: Gilles de la Tourette Syndrome; ASD: Autism Spectrum Disorder; OCD: Obsessive–Compulsive Disorder; ADHD: Attention Deficit Hyperactivity Disorder; YGTSS: Yale Global Tic Severity Scale.

**Table 3 brainsci-10-00308-t003:** Comparison between mean values of measures in each sub-phenotype and statistical significance.

	Pure GTS	GTS + OCD	GTS + ASD	GTS + ADHD	GTS + OCD + ADHD	ASD	*p*-Value
CY-BOCS	7.6 (±6)	23.5 (±8.3)	14.4 (9.3)	9.9 (±7.8)	27.1 (±1)	17.9 ± 3.1	0.0005
CY-BOCS ASD	2.2 (±1.9)	6.2 (±4.4)	17.1 (4.1)	3 (±1.9)	4.8 (±2.1)	18.1 ± 3.2	0.0005

GTS: Gilles de la Tourette Syndrome; ASD: Autism Spectrum Disorder; OCD: Obsessive–Compulsive Disorder; ADHD: Attention Deficit Hyperactivity Disorder; CY-BOCS: Children Yale Brown Obsessive–Compulsive Scale; CY-BOCS ASD: Children Yale Brown Obsessive–Compulsive Scale for Autism Spectrum Disorder.

**Table 4 brainsci-10-00308-t004:** ADOS Restricted and repetitive behaviors (D Scale): statistical significance in comparison with clinical sub-phenotypes.

	D. Total	D1	D2	D3	D4
“GTS + ASD” vs. “Pure GTS”	0.0005	0.0005	0.0005	0.0005	0.0005
“GTS + ASD” vs. “GTS + OCD”	0.0045	0.0205	0.0005	0.3828	0.9716
“GTS + ASD” vs. “GTS + ADHD”	0.0051	0.0095	0.0061	0.0065	0.0051
“GTS + ASD” vs. “GTS + OCD + ADHD”	0.0002	0.0071	0.0005	0.0005	0.7991
“ASD” vs. “Pure GTS”	0.0005	0.0005	0.0005	0.0005	0.0061
“ASD” vs. “GTS + OCD”	0.0065	0.1287	0.0005	0.6742	0.7789
“ASD” vs. “GTS + ADHD”	0.0005	0.0005	0.0054	0.0067	0.0065
“ASD” vs. “GTS + OCD + ADHD”	0.0005	0.0005	0.0005	0.0005	0.0005

GTS: Gilles de la Tourette Syndrome; ASD: Autism Spectrum Disorder; OCD: Obsessive–Compulsive Disorder; ADHD: Attention Deficit Hyperactivity Disorder. D1: unusual interests; D2: hand mannerisms and complex mannerisms; D3: repetitive and stereotyped behavior items; D4: compulsions. * represents a statistically significant *p*-value <0.05.

**Table 5 brainsci-10-00308-t005:** ADI Restricted and repetitive behaviors (C): comparison between GTS clinical sub-phenotypes.

	C Total	C1	C2	C3	C4
“GTS + ASD” vs. “Pure GTS”	0.0005	0.0005	0.0005	0.0005	0.0005
“GTS + ASD” vs. “GTS + OCD”	0.0155	0.7586	0.4183	0.1235	0.0956
“GTS + ASD” vs. “GTS + ADHD”	0.0005	0.0005	0.0335	0.0005	0.0005
“GTS + ASD” vs. “GTS + OCD + ADHD”	0.0335	0.1924	0.1639	0.001	0.0205
“ASD” vs. “Pure GTS”	0.0005	0.0005	0.0005	0.0005	0.0005
“ASD” vs. “GTS + OCD”	0.0032	0.0038	0.2365	0.0042	0.0005
“ASD” vs. “GTS + ADHD”	0.0005	0.0005	0.0005	0.0005	0.0005
“ASD” vs. “GTS + OCD + ADHD”	0.0005	0.0005	0.0005	0.0005	0.0005

C1: unusual preoccupations; C2: compulsive adherence to non-functional routines or rituals; C3: hand and finger mannerisms; C4: unusual sensory interests; GTS: Gilles de la Tourette Syndrome; ASD: Autism Spectrum Disorder; OCD: Obsessive–Compulsive Disorder; ADHD: Attention Deficit Hyperactivity Disorder. * represents a statistically significant *p* < 0.05.

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
