# Peer review of "Disentangling Restrictive and Repetitive Behaviors and Social Impairments in Children and Adolescents with Gilles de la Tourette Syndrome and Autism Spectrum Disorder"

_brainsci, 2020, doi:10.3390/brainsci10050308_

Round 1
Reviewer 1 Report
The paper is very much improved. The purpose and results are clear. My only minor recommendation is around the term "false comorbidity" and how it might be interpreted. With the current reading, it is much more clear that there is a particular group or study the paper is trying to refute (rather than a line of evidence from several groups). I know the re-organization around this concept was based on my first review and recommendation, and the paper is much easier to understand now because of the changes, but to see the term "false" associated with a specific paper and lab several times throughout the manuscript might be interpreted as confronting the group's character rather than their finding. It might interpreted by the group or by readers unfamiliar with this specific GTS research community as a bit of an attack. Other terms that might sound less confrontational might include "overlapping symptoms", "overlapping presentations" etc.
Of course, if an attack is what you wanted, then disregard this suggestion!
Author Response
Dear Reviewer,
Thanks a lot for your comments. With your suggestions the manuscript has benefited. We are very glad that now the paper is very much improved and the purposes and results are clear.
With regard to your recommendation around the term "false comorbidity" and how it might be interpreted, we want to clarify that it is not an attack to any other paper; we have just analyzed the difficulties in recognizing repetitive and restricted from compulsions in patients affected by ASD and GTS+OCD or GTS+ASD subphenotypes. This subject, sometimes, could be a challenge for the clinician and the correct diagnosis could improve the treatment and consequently the quality of life of the patients and his family. With this in mind, we would like to suggest a neuropsychological diagnostic procedure using ADI-R, ADOS, and CY-BOCS ASD in order to better distinguish symptoms.
On the other hand, we have revised again previous paper Huisman-van Dijk et al. (2016) and Darrow et al. (2017) and, even if they have studied two large clinical cohort, the used screening instrument (the Autism Quotient and the Social Responsiveness Scale), to characterize ASD symptoms not able to detect these differences. For this reason these AA stated " probable ASD in GTS" in the conclusion of the paper and suggested the need of more studies with gold standard assessment to better clarify this aspect.
With regard to English language and style we have submitted our manuscript to MDPI editing service (please, attached you could find the editing certificate).
We hope that you are satisfied with our paper
Thank you for your consideration. I look forward to hearing from you.
Best Regard
Mariangela Gulisano

Reviewer 2 Report
The authors have made significant improvements to the manuscript and have addressed my concerns. I feel that the paper does require minor editing for English style/grammar.
Author Response
Dear Reviewer,
Thanks a lot for your comments. With your suggestions the manuscript has benefited. We are very glad that now the paper is very much improved and the purposes and results are clear.
With regard to English language and style we have submitted our manuscript to MDPI editing service (please, attached you could find the editing certificate).
We hope that you are satisfied from our paper
Thank you for your consideration. I look forward to hearing from you.
Best Regard
Mariangela Gulisano

This manuscript is a resubmission of an earlier submission. The following is a list of the peer review reports and author responses from that submission.
Round 1
Reviewer 1 Report
The authors present a study investigating restrictive and repetitive behavior in children and adolescents with Gilles de la Tourette syndrome and autism spectrum disorder. There are several strengths to be noted. First, the study involves a large sample size drawn from a clinical cohort. Diagnoses were made by trained clinicians using gold standard assessments. Unfortunately, the study falls short in the analysis. I am not really sure which statistical tests the authors used because the results are not reported. The discussion section summarizes previous, relevant literature but it does not do a sufficient job of focusing on the new, relevant findings associated with the present study. I think this paper needs a major overhaul. I have listed some specific comments below but these are not exhaustive.
Method:
- The authors mention a larger ongoing longitudinal study. Have results from that study been previously published? If so they should be cited here.
- I didn’t see mention of ASD + ADHD
Results
- Significant results of statistical comparisons are mentioned without providing the actual statistical values.
Discussion:
- The first two paragraphs describe previous literature. This information would probably better fit in the Introduction. The Discussion section should be reserved for discussion of the present study results in context of previous research.
- I am really struggling to understand how the new, relevant findings from the present study fit into previous literature.
- The authors refer to a submitted study (Gulisano, et al.). I am not sure if the journal allows citing submitted work. This is important because this paper is cited a few times and seems to be a major building block for the present study.
Minor comments
- English language grammar and style edits are needed throughout the manuscript.
- Person first language (e.g. individuals with ASD) should be used throughout the manuscript.
- Line 113 - the word ‘social’ seems out of place.
Reviewer 2 Report
The idea for this article is terrific. Disentangling ASD features from GTS is a clinical conundrum for many. And the sample size and data collection methods are rigorous and using the best measures available. However, I found it hard to find the story you are trying to get across. You present the idea of a “false comorbidity” based on the overlap between repetitive behaviors in these 3 groups. In many ways your data seems to support this, but then you conclude that both groups have social problems. That and some of the analyses made the article difficult to follow, which is unfortunate because I think the topic is good and the dataset is large and strong. In addition, here are some specific things to address:
In Table 1, to which group were the patients with both ASD and GTS assigned?
Figure 1 is a helpful illustration of the CY-BOCS and CY-BOCS ASD perform in different subgroups. But I don’t fully understand whether Plots 1 and 2 add additional valuable information. Under what circumstances would the CY-BOCS ASD be used with a cutoff of 1 (or 14?).
Figure 2, about the ADOS, is a little complicated to analyze because the ADOS helped define the diagnostic groups in the first place. So, it would be expected that the groups would differ on it. But, it could still be interesting that the groups differed more on the social aspects of the ADOS than the Repetitive and Restricted behaviors. Figure 2 is depicting differences between 4 individual items of the ADOS, each with a range from 0-3 (Unusual interests, mannerisms, repetitive/stereotyped behavior, and compulsions). Are the group differences described coming from comparisons between each of these items? If so, that is a lot of comparisons on pretty limited data. The figure is interesting from a descriptive perspective, but perhaps statistical analyses should be more limited, the limitations of comparing groups on data used to define them in the first place, and the limitations of RR behavior collection on the ADOS should be described in the “Limitations” section.
The same issues with Figure 2 apply to Figure 3 about the ADI-R.
The “Discussion” section presents the false comorbidity hypothesis; if that’s what you are trying to support or refute, perhaps put it in the Introduction so the reader can follow that line of thinking when looking at the results?
The “Conclusions” section does not match your data. All your data is suggesting that RR behaviors are present in both, but that social deficits are different between groups.